



# Mapping gaseous amines, ammonia, and their particulate counterparts in marine atmospheres of China's marginal seas: Part 2 - spatiotemporal heterogeneity, causes and hypothesis

Yating Gao[1], Dihui Chen[1], Yanjie Shen[1], Yang Gao[1,2], Huiwang Gao[1,2], Xiaohong Yao[1,2*]

[1]Key Laboratory of Marine Environment and Ecology, and Frontiers Science Center for Deep Ocean Multispheres and Earth System, Ministry of Education, Ocean University of China, Qingdao 266100, China

[2]Laboratory for Marine Ecology and Environmental Science, Qingdao National Laboratory for Marine Science and Technology, Qingdao 266237, China

*$Correspondence\ to$: Xiaohong Yao (xhyao@ouc.edu.cn)

**Abstract.** In this study, spatiotemporal heterogeneities in the concentrations of alkaline gases and their particulate counterparts in the marine atmosphere over China's marginal seas were investigated in terms of causes and chemical conversion during two winter cruise campaigns, using semi-continuous measurements made by an onboard URG-9000D Ambient Ion Monitor-Ion chromatograph (AIM-IC, Thermofisher). During the cruise campaign over the East China Sea on December 27, 2019 – January 6, 2020, the concentrations of atmospheric trimethylamine ($TMA_{gas}$) varied by approximately one order of magnitude, with an average (± standard deviation) of $0.10\pm0.04$ µg m$^{-3}$ corresponding to mixing ratio of $26\pm17$ pptv. Corresponding means were $0.037\pm0.011$ µg m$^{-3}$ ($14\pm5$ pptv in mixing ratio) over the Yellow Sea on 7-16 January 2020 and $0.031\pm0.009$ µg m$^{-3}$ ($12\pm4$ pptv in mixing ratio) over the Yellow Sea and the Bohai Sea on 9-22 December 2019. In contrast, the simultaneously observed concentrations of TMA in PM$_{2.5}$, detected as TMAH$^+$, over the East China Sea were $0.098\pm0.068$ µg m$^{-3}$ and substantially smaller than $0.28\pm0.18$ µg m$^{-3}$ over the Yellow Sea and the Bohai Sea on 9-22 December 2019. A significant correlation between $TMA_{gas}$ and particulate TMAH$^+$ was obtained over the East China Sea, but no correlation existed over the Yellow Sea and Bohai Sea. The proportional or disproportional variations in concentrations of $TMA_{gas}$ with particulate TMAH$^+$ over the sea zones were likely attributed to the difference in enrichment of TMAH$^+$ in the sea surface microlayer. In addition, spatiotemporal heterogeneities in concentrations of atmospheric ammonia ($NH_{3gas}$), atmospheric dimethylamine





(DMA$_{gas}$), and DMA in PM$_{2.5}$, detected as DMAH$^+$, were also investigated. Case analyses were

performed to illustrate the formation and chemical conversion of particulate aminium ions in marine

aerosols. Finally, we hypothesized a release of basic gases and particulate counterparts from the ocean

to the atmosphere, together with secondary formation of DMAH$^+$ and chemical conversion of TMAH$^+$,

in the marine atmosphere.

**1 Introduction**

In the marine atmosphere, gaseous ammonia (NH$_{3gas}$) and amines, including trimethylamine (TMA$_{gas}$)

and dimethylamine (DMA$_{gas}$), are unique alkaline gases that play an important role in neutralizing acids

(Gibb et al., 1999; Johnson et al., 2007, 2008; Ge et al., 2011; Carpenter et al., 2012; Yu and Luo, 2014;

Paulot et al., 2015; Wentworth et al., 2016; Chen et al., 2016; Köllner et al., 2017; van Pinxteren et al.,

2019; Perraud et al., 2020). The release of NH$_{3gas}$ from the ocean to the atmosphere is determined mainly

by NH$_4^+$ concentrations in bulk seawater, surface seawater temperature, and pH of surface seawater

(Johnson et al., 2007, 2008; Carpenter et al., 2012). As organic alkali, TMA and DMA can be dissolved

in water as well as liquid organics. In addition to the factors mentioned above, the release of TMA$_{gas}$ and

DMA$_{gas}$ from the ocean to the atmosphere may also be affected by the sea surface microlayer (SML),

because of the enrichment of TMA and DMA therein (van Pinxteren et al., 2019). In addition, TMA and

DMA in bulk seawater theoretically undergo protonation as TMAH$^+$ and DMAH$^+$. However, it is unclear

whether the amines enriched in the SML undergo protonation. The differences between inorganic and

organic alkali causes different spatiotemporal variations in sea-derived emissions and concentrations of

NH$_{3gas}$ from TMA$_{gas}$ and DMA$_{gas}$, generating a large spatiotemporal heterogeneity in the molar ratios of

TMA$_{gas}$ (DMA$_{gas}$) to NH$_{3gas}$ in various marine atmospheres (Gibb et al., 1999). To explore spatiotemporal

heterogeneity and its causes, high-time-resolution observational data are required.

Two additional factors can also complicate the spatiotemporal heterogeneity of the ratios in marine

atmospheres. First, the decay of phytoplankton blooms on surface and subsurface seawater may lead to

the accumulation of NH$_4^+$ therein (Johnson et al., 2007, 2008; Liu et al., 2013). However, NH$_4^+$ is an

important nutrient and may be rapidly reused by phytoplankton in seawater (Velthuis et al., 2017; Zhang

et al., 2019a,b). The reuse of aminium ions by phytoplankton is theoretically possible, but no studies on



this have been previously reported. Two scenarios can be hypothesized: a) the reuse of aminium ions by phytoplankton as quickly as that of $NH_4^+$; and b) the slow reuse of aminium ions by phytoplankton. Second, TMA and DMA may further biochemically decompose into small molecules (Hu et al., 2015,

2018; Lidbury et al., 2014, 2015; Xie et al., 2018). These two factors would alter the ratios of $TMA_{gas}$ ($DMA_{gas}$) to $NH_{3gas}$ in oceanic emissions in opposite directions.

Unlike the release of alkaline gases, the release of primary particulate aminium aerosols from the ocean should be behaviorally similar to sea spray organic aerosols and be strongly affected by the SML (Quinn, et al., 2015; Hu et al., 2018; Dall'Osto et al., 2019). In addition to primary emissions, secondary reactions

have also been reported as important sources of particulate aminium aerosols in the marine atmosphere (Facchini et al., 2008; Müller et al., 2009; Xie et al., 2018; Hu et al., 2015, 2018; Köllner et al., 2017; Dall'Osto et al., 2019; Zhou et al., 2019). However, it is challenging to robustly identify primary aminium aerosols from secondary aminium aerosols in the marine atmosphere. Moreover, it remains poorly understood whether the detected particulate aminium ions by ion chromatography, or particulate amines

by mass spectrum, exist in the organic phase, aqueous phase, or mixed phase in the marine atmosphere (Ault et al., 2013; Prather et al., 2013; Pankow, 2015; Xie et al., 2018).

In a companion paper (Chen et al., 2021), we focused on identifying sea-derived alkali gases and particulate counterparts in $PM_{2.5}$ during a winter cruise campaign over the Yellow Sea and Bohai Sea, determined by an onboard URG-9000D Ambient Ion Monitor-Ion chromatograph (AIM-IC,

Thermofisher). In this study, we focused on investigating the spatiotemporal heterogeneity of the concentrations of $NH_{3gas}$, $TMA_{gas}$, and $DMA_{gas}$, together with their particulate counterparts in marine atmospheres, by comparing observations during two winter cruise campaigns over the Yellow Sea, Bohai Sea, and the East China Sea. Moreover, previously reported episodic concentrations of particulate $TMAH^+$ and $DMAH^+$ observed in the marine atmosphere over the Yellow Sea were also included to

deepen our understanding of the size distributions of the aminium ions and the ratio of aminium ions to $NH_4^+$ and related primary or secondary origins of particulate aminium ions. Building on the analysis results, a hypothesis is presented to illustrate the release of gaseous alkali and their counterparts from the ocean to the atmosphere, and related chemical conversions in the marine atmosphere.



## 2 Experimental

From December 27, 2019, to January 17, 2020, a round cruise survey, focusing on air/sea exchanges of greenhouse gases and short-life reactive gases, was conducted over the East China Sea and the Yellow Sea, in China, using an R/V Dongfanghong-3. The cruise routes during the campaign and immediately before are shown in Figure S1a,b. The cruise campaigns on 9-22 December 2019 and December 27, 2019 – January 17, 2020, are referred to as Campaign A and B, respectively, in this study. Details on the

measurements during Campaign B were the same as those reported in the companion paper, that is, the onboard AIM-IC was housed in air-conditioned containers and measured the concentrations of gaseous species of interest, and particulate counterparts, in $PM_{2.5}$. In Campaign B, no $K^+$ contamination occurred in the channel used to determine gaseous species, and the concentrations of $DMA_{gas}$ and $TMA_{gas}$ could be determined properly (Fig. 1a). However, strong $K^+$ contamination unexpectedly occurred in the

channel used to determine particulate species from January 7, 2020, leading to no data for $DMAH^+$ and $TMAH^+$ in $PM_{2.5}$, after that date (Fig. 1b). However, the concentrations of $NH_4^+$ and other ions, excluding $K^+$, were not affected because their residence time in the ion chromatograph was far away from that of $K^+$.

The AIM-IC expectedly encountered terminations several times during Campaign B. This is quite normal

for most online analyzers operating after two weeks on a swaying research vessel, especially when the cruise frequently encounters strong winds. Considering that strong winds substantially increase air/sea exchange fluxes, all instruments were operated to continuously capture the signals. After restarting the AIM-IC, it always reported a few abnormally high values in the first 3-5 h because of residuals in the system. Abnormal values were excluded from the analysis.

In addition, observations made over the Yellow Sea on 2-21 May 2012 were also included to facilitate analyses. These data have been reported in our previous study (Hu et al., 2015), in which the total concentrations of $TMAH^+$ in three size-segregated atmospheric particle samples can also reach a high level of ~1 μg m$^{-3}$. Note that high concentrations of particulate $TMAH^+$ were not observed in marine atmospheres during additional multiple cruise campaigns from the marginal seas of China to the

northwest Pacific Ocean (Xie et al., 2018; Hu et al., 2018; Zhu et al., 2019). In the study reported by Hu et al. (2015), a low-volume Anderson cascade impactor (AN-200; Sibata Co., Inc., Japan) was employed



to collect atmospheric particles with 50% aerodynamic cut-off diameters of 11, 7.0, 4.7, 3.3, 2.1, 1.1, 0.65, and 0.43 µm. Details of the sampling and chemical analyses can be found in Hu et al. (2015). The cruise campaign was referred to as Campaign C in this study, and the sea zones collected from the three

aerosol samples are shown in Figure S1c.

## 3 Results and discussion

### 3.1 Spatiotemporal variations in concentrations of alkaline gases over the East China Sea and the Yellow Sea

Figure 1a,b shows spatiotemporal variations in concentrations of $TMA_{gas}$, $DMA_{gas}$, and $NH_{3gas}$ and their

counterparts in $PM_{2.5}$ during Campaign B. The corresponding wind speeds and directions are shown in Figure 1c. Some concentrations of $TMA_{gas}$, particulate $TMAH^+$, and wind fields are also mapped in Figure 1d-f. The concentrations of $TMA_{gas}$ ranged from 0.022 µg m$^{-3}$ (8 pptv in mixing ratio) to 0.22 µg m$^{-3}$ (91 pptv in mixing ratio) over the East China Sea on December 27, 2019 – January 6, 2020. Corresponding average values were 0.10±0.04 µg m$^{-3}$ (26±17 pptv in mixing ratio).

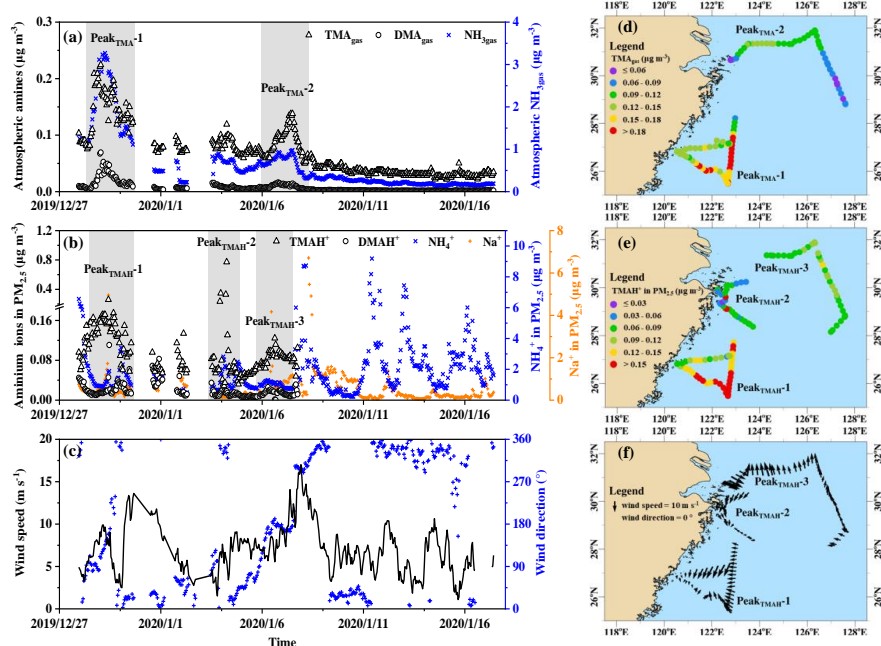




**Figure 1: Time series and maps of basic gases and particulate counterparts in concentration and meteorological parameters during the cruise campaign on 27 December 2019 to 17 January 2020 (time series of $TMA_{gas}$, $DMA_{gas}$ and $NH_{3gas}$ (a); time series of $TMAH^+$, $DMAH^+$ and $NH_4^+$ in $PM_{2.5}$ (b); time series of wind speed and wind directions (c), map of $TMA_{gas}$ (d); map of $TMAH^+$ (e); map of wind fields (f); not all data were shown in (d-f) to avoid clustering.**


The values largely decreased to $0.037\pm0.011$ µg m$^{-3}$ ($14\pm5$ pptv in mixing ratio) over the Yellow Sea on 7-16 January 2020. The latter concentrations were comparable to those of $0.031\pm0.009$ µg m$^{-3}$ ($12\pm4$ pptv in mixing ratio) observed over the Yellow Sea and the Bohai Sea during Campaign A (Chen et al., 2021). Based on the evidence provided below, the observed $TMA_{gas}$ during the period of Campaign B

was probably determined by the actual time emission potentials of $TMA_{gas}$ from the cruise sea zone. Long-range continental transport should be a negligible contributor to the observed $TMA_{gas}$ in the marine atmosphere.

A moderately good exponential correlation, ($TMA_{gas}=0.03\times e^{0.08T}$; $R^2=0.76$, P<0.01), was demonstrated between the concentrations of $TMA_{gas}$ and ambient air temperature (Fig. 2a). Although the surface

seawater temperature was not measured, it can reasonably be approximated from ambient air temperature (Deng et al., 2014). The exponential correlation suggested that the observed concentrations of $TMA_{gas}$ were probably determined by the emission potentials of $TMA_{gas}$ at the same time, in the corresponding sea zones. Across the same ambient temperature ranges, the observed concentrations of $TMA_{gas}$ over the East China Sea (full dots in Fig. 2a) were larger than those over the Yellow Sea (empty dots in Fig. 2a).

The regression equation derived was $TMA_{gas}=0.03\times e^{0.05T}$ ($R^2=0.56$, P<0.01) when data measured over the Yellow Sea during Campaign B were used alone. By comparing the two derived regression equations, it can be inferred that the actual time emission potentials of $TMA_{gas}$ over the East China Sea were larger than those over the Yellow Sea. Considering approximately constant pH values of 8.0-8.2 in surface seawater across the two sea zones (Lui et al., 2015; Shao et al., 2020), the concentrations of $TMAH^+$ in

the surface seawater of the East China Sea were expected to be larger than those over the Yellow Sea during Campaign B. Unfortunately, no direct measurements were made to confirm this.

To enlarge the data set measured over the Yellow Sea, measurements made at 15:00 on December 16 – 01:00 on December 19 during Campaign A were included. During this period in Campaign A, the concentrations of $TMA_{gas}$ were higher than those observed during other periods in Campaign A at the

same ambient air temperature (Chen et al., 2021). We combined the data during this period with data



measured over the Yellow Sea during Campaign B to derive the regression equation: $TMA_{gas}=0.03\times e^{0.05T}$ (Fig. S2), which is the same as that derived from the data measured over the Yellow Sea during Campaign B alone. However, $R^2$ slightly decreased to 0.54, with $P<0.01$. This result further supports the lower actual time emission potentials of $TMA_{gas}$ from the Yellow Sea.

The concentrations of $TMA_{gas}$ in the continental atmosphere upwind of the Yellow Sea during the summer of 2019 remained at a low level of ~0.002 µg m$^{-3}$ (Chen et al., 2021) and were over one order of magnitude smaller than the values over the Yellow Sea on 7-16 January 2020. An even larger difference existed when the observed concentrations of $TMA_{gas}$ over East China were compared with continental values. Unfortunately, no recent measurements of $TMA_{gas}$ in the continental atmosphere upwind of the

East China Sea were available for comparison.

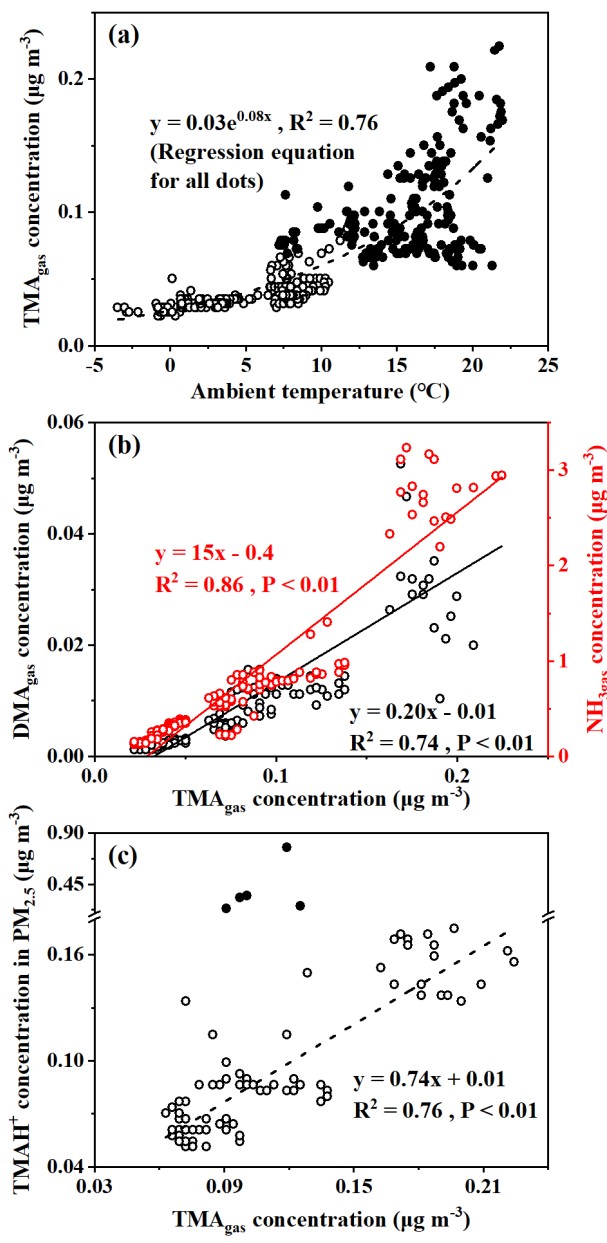

**Figure 2: Correlations of TMA$_{gas}$ with ambient air temperature, DMA$_{gas}$, NH$_{3gas}$ and TMAH$^+$ in PM$_{2.5}$ with TMA$_{gas}$ (TMA$_{gas}$ versus ambient air temperature (a); DMA$_{gas}$ and NH$_{3gas}$ versus TMA$_{gas}$ (b); TMAH$^+$ versus**



TMA$_{gas}$ (b); full dots in (b) represent five episodic concentrations of TMAH$^+$ and were excluded for
correlation analysis)

No increase in TMA$_{gas}$ were detected with several periodically large increases in particulate NH$_4^+$ under

offshore winds over the Yellow Sea on 7-16 January 2020 (Fig. 1a-c). In contrast, higher concentrations

of NH$_4^+$ were associated with lower values of TMA$_{gas}$ over the East China Sea, and vice versa (Fig. 1a,

b; the start period of Campaign B). Moreover, two broad peaks of TMA$_{gas}$ were observed over the East

China Sea, approximately 200 km from the continent, under onshore winds (Fig. 1d, f). Combining

concentrations of TMA$_{gas}$ in the continental atmosphere upwind of the Yellow Sea with these results

allowed us to infer that continental transport represents a negligible contribution to the observed TMA$_{gas}$

during Campaign B.

Spatiotemporal variations in concentrations of DMA$_{gas}$ and NH$_{3gas}$ were similar to those of TMA$_{gas}$ during

Campaign B. For example, the concentrations of DMA$_{gas}$ and NH$_{3gas}$ varied 0.012±0.011 µg m$^{-3}$ and

1.1±0.76 µg m$^{-3}$, respectively, over the East China Sea. However, they largely decreased to 0.002±0.001

µg m$^{-3}$ and 0.24±0.08 µg m$^{-3}$, respectively, over the Yellow Sea. In addition, the concentrations of

DMA$_{gas}$ and NH$_{3gas}$ had moderately good and good correlations with those of TMA$_{gas}$ (Fig. 2b): DMA$_{gas}$

=0.20×[TMA$_{gas}$] -0.01, R$^2$=0.74, P<0.01, and NH$_{3gas}$ =15×[TMA$_{gas}$] -0.40, R$^2$=0.86, and P<0.01. The

correlations suggested that the observed DMA$_{gas}$ and NH$_{3gas}$ were also generally derived from marine

emissions simultaneously with TMA$_{gas}$. Thus, we conclude that the seas were the net source of DMA$_{gas}$

and NH$_{3gas}$ during the study. Note that the observed ratios of TMA$_{gas}$ to NH$_{3gas}$ were two orders of

magnitude larger than those previously reported in marine atmospheres and adopted for modeling (Van

Neste, et al., 1987; Gibb et al., 1999; Yu and Luo, 2014). However, the observed ratios of DMA$_{gas}$ to

NH$_{3gas}$ were reasonably comparable to values previously reported (Yu and Luo, 2014).

### 3.2 Spatiotemporal variations in concentrations of particulate TMAH$^+$, DMAH$^+$ and NH$_4^+$ over the East China Sea

The concentrations of TMAH$^+$ in PM$_{2.5}$ varied around 0.098±0.068 µg m$^{-3}$ over the East China Sea, but

no data could be obtained over the Yellow Sea during the cruise because of K$^+$ contamination. Almost

all values were smaller than 0.2 µg m$^{-3}$, except five episodic values of 0.26 µg m$^{-3}$ at 10:00 on 29

December 2019, 0.23 µg m$^{-3}$ and 0.35 µg m$^{-3}$ at 22:00-23:59 on 3 January and 0.33 µg m$^{-3}$ and 0.77 µg





$m^{-3}$ at 05:00-06:59 on 4 January 2020 (Fig. 1b). The concentrations of $TMAH^+$ exhibited a moderately good correlation with those of $TMA_{gas}$ simultaneously observed over the East China Sea when the five episodes with concentrations of $TMAH^+$ in $PM_{2.5}$ exceeding 0.2 μg $m^{-3}$ were excluded for correlation

(Fig. 2c), suggesting that the $TMAH^+$ in $PM_{2.5}$ may also be derived from marine sources. In addition, a broad peak of $TMAH^+$ concentrations ($Peak_{TMAH-1}$ shadowing in Fig. 1b) was observed on 27-30 December 2019, when a negative correlation existed between the concentrations of $TMAH^+$ and $NH_4^+$, with $R^2=0.35$, and $P<0.01$. The negative correlation also supported the conclusion that increased concentrations of $TMAH^+$ in $PM_{2.5}$ were driven by enhanced marine emissions rather than continental

transport.

The large increases in concentrations of particulate $NH_4^+$, for example, when its concentration exceed 5 μg $m^{-3}$, under offshore winds, clearly indicated the continental transport of air pollutants (Figs. 1bc, S1a). However, when its concentration was below 1 μg $m^{-3}$, a significant correlation between particulate $NH_4^+$ and $TMAH^+$ was apparent, with $P<0.01$ (empty dots Fig. 3a). When five points with concentrations of

particulate $TMAH^+$ exceeding 0.2 μg $m^{-3}$ were included in the correlation analysis (full dots in Fig. 3a), the $R^2$ increased to 0.62. Thus, the primary sea-derived particulate $NH_4^+$ could not be excluded in the marine atmosphere over the East China Sea. On the basis of the regression equation shown in Figure 3a, the estimated primary sea-derived particulate $NH_4^+$ should be smaller than 0.48 μg $m^{-3}$ under concentrations of particulate $TMAH^+$ below 0.2 μg $m^{-3}$. Altieri et al. (2014) used isotopic data and

identified a marine ammonium source in rainwater in Bermuda, but they did not specify whether the marine ammonium was derived from the primary particulate emission.

The concentrations of $DMAH^+$ in $PM_{2.5}$ varied around 0.019±0.014 μg $m^{-3}$ over the East China Sea. The average value was only one-fifth that of $TMAH^+$ in $PM_{2.5}$, but it was almost double that of $DMA_{gas}$ simultaneously observed. The average value of $DMAH^+$ in $PM_{2.5}$ was also approximately one-third of

the value observed over the Yellow Sea and the Bohai Sea on 9-22 December (0.065±0.068 μg $m^{-3}$) (Chen at al., 2021). Positive correlations between $DMAH^+$ and $NH_4^+$ were demonstrated, with $P<0.01$, but the $R^2$ value was 0.17 (all dots in Fig. 3b). However, when $NH_4^+$ concentrations exceeded 5 μg $m^{-3}$, there was a good correlation between $DMAH^+$ and $NH_4^+$ ($DMAH^+ = 0.014×[NH_4^+]$ - 0.049, $R^2=0.80$, $P<0.01$) (full dots in Fig. 3b). When $NH_4^+$ concentrations were in the range 2-4 μg $m^{-3}$ (half full dots in

Fig. 3b), a moderately good correlation of $DMAH^+$ existed with $NH_4^+$ ($DMAH^+ = 0.013×[NH_4^+]$ - 0.012,





$R^2$=0.71, P<0.01), when three outliers were omitted. The good and moderately good correlations, together with the negative intercepts in the regression equations, suggested a dominant contribution from continental transport to the observed $DMAH^+$ when $NH_4^+$ concentrations exceeded 2 μg m$^{-3}$, except for the three outliers.

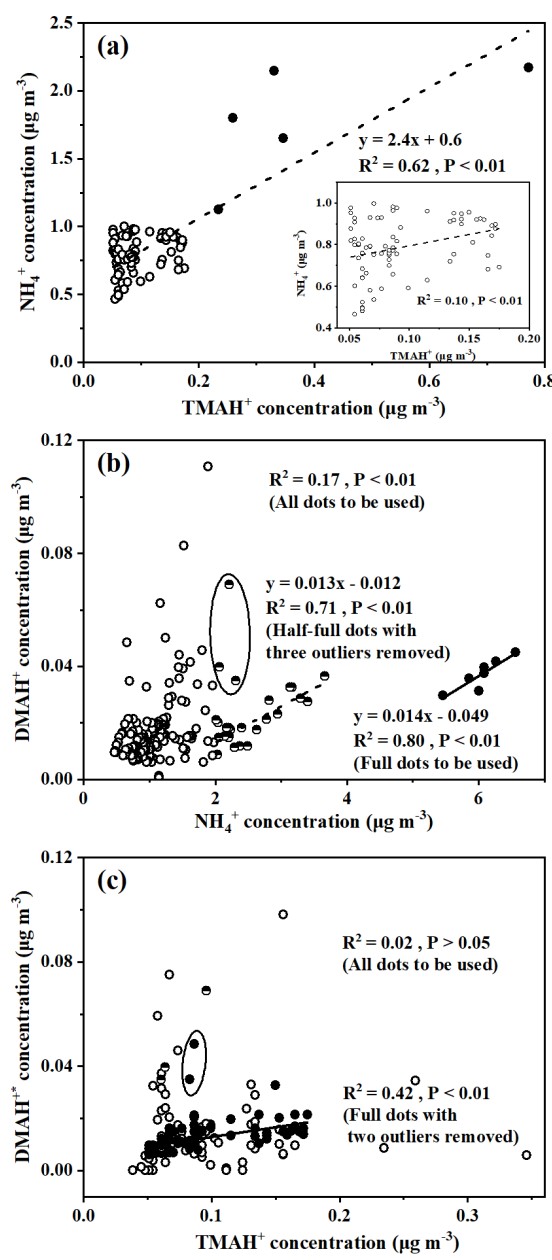


**Figure 3: Correlations between concentrations of ions in PM$_{2.5}$ NH$_4^+$ versus TMAH$^+$ (a) DMAH$^+$ versus NH$_4^+$; (b) DMAH$^{+*}$ versus NH$_4^+$; (c) DMAH$^{+*}$ was defined in the text; full, half full and empty dots in (a), (b) and (c) are defined in the text.**



When the secondary regression equation, with the concentrations of $NH_4^+$ ranging from 1 µg m$^{-3}$ to 2 µg

m$^{-3}$ as input, was used to estimate the concentrations of $DMAH^+$ from continental transport, the estimated

concentrations accounted for 33±27% of the observed values. The sea-derived $DMAH^+$ in $PM_{2.5}$ was

likely the major contributor to the observed values in most cases. In the three outliers having

concentrations of particulate $NH_4^+$ between 2.1 and 2.3 µg m$^{-3}$ (half full dots in Fig. 3b), the contributions

from continental transport were estimated to be 24%, 37% and 52%, respectively. When the

concentrations of $NH_4^+$ were smaller than 1 µg m$^{-3}$, the values predicted by the secondary regression

equation were close to or smaller than zero. Under such low concentrations of $NH_4^+$, the sea-derived

particulate $NH_4^+$ may contribute appreciably to the observed value. The estimated percentages are the

upper values. Thus, the observed $DMAH^+$ in $PM_{2.5}$, when $NH_4^+$ concentrations were below 1 µg m$^{-3}$,

should be overwhelmed by marine sources. Under these conditions, a significant correlation with a low

$R^2$ was obtained between $DMAH^+$ and $TMAH^+$ when two outliers were removed (full dotes in Fig. 3c,

$R^2$=0.42, P<0.01). However, primary emissions of particulate $DMAH^+$ from the East China Sea likely

acted as a minor contributor to the observed values. For example, considering four of the five episodic

concentrations of particulate $TMAH^+$ ranging from 0.23 µg m$^{-3}$ to 0.77 µg m$^{-3}$, the corresponding

concentrations of particulate $DMAH^+$ varied from 0.011 µg m$^{-3}$ to 0.018 µg m$^{-3}$. Assuming that the net

increase in particulate $DMAH^+$ with increasing particulate $TMAH^+$ was derived from primary emissions,

the concentrations of the primary sea-derived particulate $DMAH^+$ were estimated to be smaller than 0.003

µg m$^{-3}$ when particulate $TMAH^+$ concentrations were below 0.2 µg m$^{-3}$. The campaign average of

particulate $DMAH^+$ was 0.019 µg m$^{-3}$. In contrast, the higher value of particulate $DMAH^+$ in the sample

with the episodic concentration of particulate $TMAH^+$ at 0.26 µg m$^{-3}$, was 0.046 µg m$^{-3}$, which may be

related to secondary formation of $DMAH^+$. The secondary formation of $DMAH^+$ in the marine

atmosphere was also speculated to be the major source of the observed $DMAH^+$ during most periods.

### 3.3 In-depth analysis during three episodes

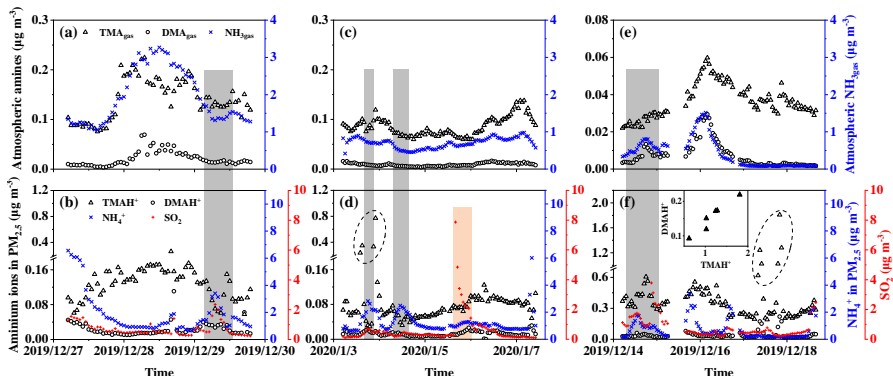

**Figure 4: Times series of concentrations of gases and particulate ions during three episodes. Basic gases in**

**E-period 1 (a); particulate ions and $SO_2$ in E-period 1 (b); (c) and (d) same as (a) and (b) except in E-period 2; (e) and (f) same as (a) and (b) except in E-period 3; grey and pink shadowing represent episodes with increasing $NH_4^+$ or $SO_2$, respectively; Fig superimposed in (f) show the correlation between $TMAH^+$ and $DMAH^+$ in six cycling points in (f))**

Three episodes were further selected for deeper analyses of the sea-derived alkaline gases and primary

particulate counterparts, during which continental transport was likely to have largely decreased. E-period 1 started on 23:00 on December 27 to 13:00 on December 30, 2019, when increases in concentrations of sea-derived gases and sea-derived primary $TMAH^+$ in $PM_{2.5}$ were observed. E-period 2 started on 13:00 on January 3 to 18:00 on January 7, 2020, when 1) an episodic increase in the sea-derived primary $TMAH^+$ in $PM_{2.5}$ occurred in the absence of a corresponding increase in $TMA_{gas}$; 2) an

increase in the concentration of sea-derived $TMA_{gas}$ was observed without a corresponding increase in sea-derived primary $TMAH^+$ in $PM_{2.5}$. Both E-period 1 and E-period 2 were observed over the East China Sea during Campaign B. E-period 3 started on 00:00 on 15 December 11:00 on 19 December 2019 during Campaign A, when either an increase in concentration of $TMA_{gas}$ or particulate $TMAH^+$ was observed without a corresponding increase in their counterparts, similar to E-period 2.

The concentrations of $TMAH^+$ in $PM_{2.5}$ during E-periods 1 and 2 were smaller than those during period 3, and the reverse was generally true for the concentrations of $TMA_{gas}$. This was true of all observations over the East China Sea during Campaign B, in comparison with those measured during Campaign A. For example, the average concentration of $TMAH^+$ in $PM_{2.5}$ during Campaign A was 0.28 μg m$^{-3}$ (Chen


et al., 2021), which was approximately three times the corresponding average of 0.098 µg m$^{-3}$ during

Campaign B.

The concentrations of $TMA_{gas}$ and $TMAH^+$ in $PM_{2.5}$ were generally comparable during periods E-periods 1 and 2. However, the concentrations of $TMA_{gas}$ were approximately one order of magnitude smaller than those of $TMAH^+$ in $PM_{2.5}$ during E-period 3. The large difference between $TMA_{gas}$ and particulate $TMAH^+$ were observed over the Yellow Sea and Bohai Sea throughout Campaign A. Several factors,

including surface seawater temperature, sea surface wind speed, and the concentration of $TMAH^+$ in surface seawater and/or the SML, among others, may cause the disproportion, as discussed below.

As analyzed earlier, higher surface seawater temperatures, together with possibly higher concentrations of $TMAH^+$ in surface seawater, likely increased the concentrations of $TMA_{gas}$ over the East China Sea, relative to those over the Yellow Sea and Bohai Sea. However, these two factors could not explain the

lower concentrations of $TMAH^+$ in $PM_{2.5}$ over the East China Sea compared to concentrations over the Yellow Sea and Bohai Sea. The release of sea spray aerosols is generally an exponential function of wind speed (Andreas, 1998; Leeuw et al., 2011; Feng et al., 2017). Thus, sea surface wind speeds are now examined. Hourly average wind speeds were 7.3±2.6 m s$^{-1}$ over the East China Sea during Campaign B, which were not significantly different from those of 7.9±8.1 m s$^{-1}$ during Campaign A (P>0.05).

Moreover, five hourly averages of $TMAH^+$ in $PM_{2.5}$ exceeded 1 µg m$^{-3}$ over the Yellow Sea and Bohai Sea when wind speeds reached 12±0.5 m s$^{-1}$. During the nine hourly average wind speeds exceeding 12 m s$^{-1}$ during the East China Sea cruise, the corresponding concentrations of $TMAH^+$ in $PM_{2.5}$ were only 0.08±0.01 µg m$^{-3}$. Five concentrations of $TMAH^+$ in $PM_{2.5}$ exceeded 0.2 µg m$^{-3}$ in Campaign B and wind speeds ranged from 5.6 to 8.1 m s$^{-1}$ at those moments. Therefore, wind speeds alone were unable

to explain the observed lower concentrations of $TMAH^+$ in $PM_{2.5}$ over the East China Sea, compared to concentrations over the Yellow Sea and Bohai Sea.

Because the SML affects all mass transfers between the atmosphere and ocean (Cunliffe et al., 2013; Quinn, et al., 2015), the release of sea-spray aerosols containing $TMAH^+$ should be affected by the abundance of $TMAH^+$ in SML, in addition to sea surface wind speeds and concentrations of $TMAH^+$ in

bulk surface seawater. Combining the observational facts mentioned above, we argue that $TMAH^+$ may be more highly enriched in the SML than in bulk surface seawater over the Yellow Sea and Bohai Sea,



during Campaign A under the low surface seawater temperatures. Direct measurements of TMAH$^+$ enriched in the SML, as reported by van Pinxteren et al. (2019), are needed to confirm this hypothesis.

During E-period 1, the concentrations of TMA$_{gas}$ and DMA$_{gas}$ exhibited similar spatiotemporal patterns.

The concentrations of NH$_{3gas}$ exhibited a spatiotemporal pattern similar to that of gaseous amines during the initial period of increasing concentrations and the late period of decreasing concentrations, but during the transition between early and late periods. The imbalance between implied varying ratios of aminium over ammonium in bulk surface seawater and/or the SML of the corresponding sea zone. The concentrations of particulate TMAH$^+$ exhibited a spatiotemporal pattern similar to that of gaseous amines,

while the reverse spatiotemporal pattern was found for concentrations of particulate DMAH$^+$. The primary sea-spray aerosols may contain substantially low concentrations of particulate DMAH$^+$, as mentioned above. In addition, a significant decrease in the concentration of particulate TMAH$^+$ was apparent with increasing concentrations of particulate NH$_4^+$ and DMAH$^+$, as well as those of SO$_2$ (grey shadowing in Fig. 4a). The unique decrease in particulate TMAH$^+$ also occurred in E-period 2 and E-

period 3 (grey and pink shadowing in Fig. 4d,f), regardless of the simultaneous increase or decrease in concentrations of TMA$_{gas}$. Secondary chemical reactions likely converted particulate TMAH$^+$ to compounds that were undetectable by AIM-IC.

Unlike during E-Period 1, the disproportional release of TMA$_{gas}$ with particulate TMAH$^+$ from the seas likely occurred in E-periods 2 and 3. Moreover, a large increase in the concentration of particulate

DMAH$^+$ was observed simultaneously with a large increase in particulate TMAH$^+$ in the six episodes observed over the Yellow Sea (Figure superimposed in Fig. 4f). However, only a small increase in particulate DMAH$^+$ was detected for the four episodes observed over the East China Sea (cycled empty triangles in Fig. 4d). This disproportion may also be ascribed to the spatiotemporal heterogeneity of enrichments of TMAH$^+$ and DMAH$^+$ in the SML.

**3.4 Molar ratios of gaseous amines over NH$_{3gas}$ and their particulate counterparts**

The dissociation constants (K$_b$) of TMA and DMA in water were 31 and 4 times that of NH$_3$•H$_2$O (Ge et al., 2011), respectively. Thus, DMA$_{gas}$ and TMA$_{gas}$ may enable the competitive neutralization of acids by NH$_{3gas}$ in the atmosphere (Almeida et al., 2013; Chen et al., 2016; Yao et al., 2018; Xie et al., 2018). When the values of K$_b$ were used to calculate the effective Henry's Law constants for DMA ($^{eff}$KDMA),





TMA ($^{eff}$KTMA), and NH$_3$ ($^{eff}$KNH$_3$), assuming activity coefficients to be unity, the ratios of

$^{eff}$KDMA/$^{eff}$KNH$_3$ and $^{eff}$KTMA/$^{eff}$KNH$_3$ were 16 and 0.6, respectively, at an ambient temperature of 298

K under acidic conditions (Ge et al., 2011). Considering the large differences between $^{eff}$KDMA/$^{eff}$KNH$_3$

and $^{eff}$KTMA/$^{eff}$KNH$_3$, the molar ratios of TMA$_{gas}$ over NH$_{3gas}$ and the ratios of DMA$_{gas}$ to NH$_{3gas}$ were

separately examined.

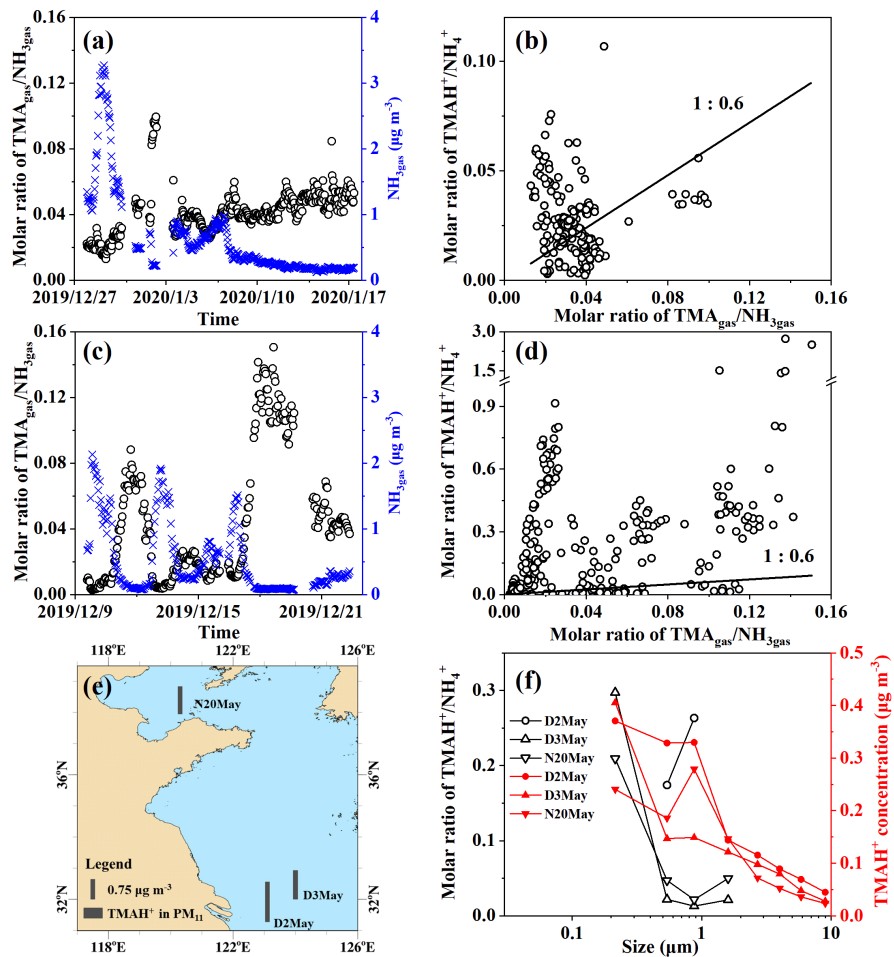


**Figure 5: Time series of molar ratios of TMA$_{gas}$/NH$_{3gas}$ (a) and (c) in Campaign B and A; correlation between TMAgas/NH$_{3gas}$ and TMAH$^+$/NH$_4^+$ (b) and (d) in Campaign B and A;, map of particulate TMAH$^+$ (e) and size distributions of TMAH$^+$/NH$_4^+$ and mass concentrations of TMAH$^+$ (f) in Campaign C.**





The ratios were first examined during Campaign B, when higher concentrations of $TMA_{gas}$ and $DMA_{gas}$

were observed. A large spatiotemporal variation in the molar ratio of $TMA_{gas}$ to $NH_{3gas}$, ranging between

0.013 to 0.10 over the East China Sea, was observed on December 27, 2019 – January 7, 2020 (Fig. 5a).

Low ratios with a mean of 0.022±0.004 occurred concurrently with higher concentrations of $TMA_{gas}$ and

$NH_{3gas}$, for example, from 23:00 on December 27, 2019, to 13:00 on December 30, 2019 (Peak$_{TMA-1}$ in

Fig. 1a). Increased ratios of 0.08-0.10 occurred concurrently with the lowest concentrations of $NH_{3gas}$,

ranging between 0.22 and 0.28 μg m$^{-3}$ from 22:00 on 1 January and 07:00 on 2 January 2020. This

phenomenon may be related to the reuse of $NH_4^+$ by phytoplankton (Liu et al., 2013). In Campaign B

over the Yellow Sea on 7-17 January 2020, the ratios exhibited a narrow range of 0.034 to 0.064; one

outlier of 0.085 was excluded (Fig. 5a).

During Campaign A over the Yellow Sea and Bohai Sea on 9-22 December, the molar ratios $TMA_{gas}$ to

$NH_{3gas}$ ranged from 0.003 to 0.15 (Fig. 5c). The ratios increased during the period from 17:00 on

December 17 to 16:00 on December 19, with a mean of 0.12±0.014, because of a large decrease in the

concentrations of $NH_{3gas}$ (Figs. 5c and 4e). However, smaller ratios in the range of 0.011-0.016 were

observed between 20:00 on December 16 to 00:00 on December 17 in the presence of the strong sea-

derived emissions of alkaline gases (Figs. 5c and 4e). These results were consistent with those observed

in Campaign B, indicating that the ratios of $TMA_{gas}$ to $NH_{3gas}$ during periods of episodic emission were

likely decreased by half to one order of magnitude relative to those during periods of low emission.

The mean molar ratio of $TMAH^+$ to $NH_4^+$ in $PM_{2.5}$ was 0.032±0.019 during Campaign B over the East

China Sea, comparable to those of $TMA_{gas}$ to $NH_{3gas}$ (Fig. 5c). When the molar ratios of $TMAH^+$ to $NH_4^+$

in $PM_{2.5}$ were plotted against the ratios of $TMA_{gas}$ to $NH_{3gas}$, data were scattered along the 1:0.6 line.

However, no significant correlation was observed between them. The observed particulate $TMAH^+$ may

co-exist externally with aerosols containing $NH_4^+$.

During Campaign A, the molar ratios of $TMAH^+$ to $NH_4^+$ largely varied with the 25th, 50th, 75th, and

90th percentile values of 0.009, 0.089, 0.35, and 0.56, respectively. As extremes, the 98th-100th

percentile values ranged between 1.4 and 2.7, when concentrations of $TMAH^+$ in $PM_{2.5}$ exceeded 1 μg

m$^{-3}$. When the molar ratios of $TMAH^+$ to $NH_4^+$ in $PM_{2.5}$ were plotted against the ratios of $TMA_{gas}$ to

$NH_{3gas}$ (Fig. 5d), no significant correlation was apparent and most of these data were distributed far above

the 1:0.6 line. To explain these results (Pankow, 2015; Xie et al., 2018), laboratory experiments are





required to measure the thermodynamic gas-aerosol equilibria in the organic phase. Although the particulate TMA was detected as $TMAH^+$ by AIM-IC, it may not necessarily occur protonated in sea-

spray organic aerosols.

Measurements of ions' concentrations in $PM_{2.5}$ do not demonstrate the size distributions of the ratios of $TMAH^+$ to $NH_4^+$. Thus, three episodes, with concentrations of total particulate $TMAH^+$ exceeding 1 µg $m^{-3}$ in atmospheric particles with diameter smaller than 11 µm ($PM_{11}$) collected over the Yellow Sea in 2012 (Hu et al., 2015), were included in the analysis. The sample collection sea zones are mapped in

Figure 5e. The size distributions of particulate $TMAH^+$ in mass concentration and molar ratios of $TMAH^+$ to $NH_4^+$ are shown in Figure 5f.

The concentrations of $TMAH^+$ generally increased from the bin-size of 7.0-11 µm to that of <0.43 µm (Fig. 1f), which were totally different from those of $NH_4^+$, which peaked at 0.65-1.1 µm (Figure was superimposed in Fig. S1c). The unique size distributions of particulate $TMAH^+$ also implied that the

observed $TMAH^+$ was overwhelmingly derived from primary sea spray organic aerosols, based on laboratory experimental results and field measurements (Ault et al., 2013; Prather et al., 2013; Hu et al., 2015, 2018; Quinn et al., 2015). Note that mass concentration size distribution patterns of particulate $TMAH^+$ were reported similar to those of $NH_4^+$ when secondary-formed particulate $TMAH^+$ dominated the primary particulate $TMAH^+$ (Hu et al., 2018; Xie et al., 2018).

The ratios of $TMAH^+$ to $NH_4^+$ in bins of different sizes were also calculated. Assuming 1) gas-aerosol equilibria had been achieved and particulate $TMAH^+$ to $NH_4^+$ co-existed internally, the ratios in different-sized particles should theoretically approach a constant. However, the ratios in particle size bins were distributed across two different ranges, namely 0.2-0.3 and 0.01-0.05, corresponding to concentrations of $NH_4^+$ exceeding 0.9 µg $m^{-3}$, or below 0.6 µg $m^{-3}$, respectively, rejecting the null hypothesis. Note that

the ratios were not calculated in size bins when the concentrations of $NH_4^+$ were smaller than 0.1 µg $m^{-3}$. At such low concentrations, the analytic errors may be large and can be transferred to the calculated ratios.

The time series of ratios of $DMA_{gas}$ to $NH_{3gas}$, particulate $DMAH^+$ to particulate $NH_4^+$, and their correlations during Campaign A and B are shown Figure S3a,b,c,d. The concentrations of $DMAH^+$ in the

three episodic samples collected in 2012 are mapped in Figure S3e. Size distributions of particulate $DMAH^+$ in mass concentration and molar ratios of $DMAH^+$ to $NH_4^+$ are shown in Figure S3f. During





Campaigns B and A, the mean molar ratios of $DMA_{gas}$ to $NH_{3gas}$ was 0.004±0.002 and 0.006±0.005, respectively. When the molar ratios of $DMAH^+$ to $NH_4^+$ in $PM_{2.5}$ were plotted against the ratios of $DMA_{gas}$ over $NH_{3gas}$ (Fig. 5d), the data were far below the 1:16 line during Camping B. A possible

explanation was that the sea-derived $DMA_{gas}$ was not achieved with the $NH_4^+$- containing aerosols from continental transport. During Campaign A, most of the data were also far below the 1:16 line. However, there were a few points close to, or above, the 1:16 line. The data were associated with the strong sea-derived primary particulate $DMAH^+$, which may co-exist externally with $NH_4^+$- containing aerosols. In addition, the size distributions of particulate $DMAH^+$ in the mass concentration and molar ratios of

$DMAH^+$ to $NH_4^+$ in the three samples collected in 2012 were generally similar to those of $TMAH^+$. The analysis of particulate $TMAH^+$ was applied to that of particulate $DMAH^+$.

**4 Conclusions and hypotheses**

Semi-continuous measurements of concentrations of basic gases and their counterparts over the East China Sea, Yellow Sea, and Bohai Sea showed large spatiotemporal variations. The average

concentration of $TMA_{gas}$ over the East China Sea in Campaign B was 0.1 μg m$^{-3}$, but decreased by approximately 70% over the Yellow Sea and Bohai Sea in Campaigns A and B. In contrast, the average concentration of $TMAH^+$ in $PM_{2.5}$, over the East China Sea was 0.098 μg m$^{-3}$, while the average increased by approximately 200% in Campaign A. Comprehensive analysis indicated that both $TMA_{gas}$ and particulate $TMAH^+$ were released from the seas. The disproportional release of $TMA_{gas}$ and particulate

$TMAH^+$ from the East China Sea, compared with the Yellow Sea and the Bohai Sea, however, pointed to a differential enrichment of $TMAH^+$ in the SML. We hypothesized that lower surface seawater temperature would reduce the rate of biochemical degradation of polysaccharides, peptides, and protein gels (Carpenter et al., 2012; Prather et al., 2013; Quinn et al., 2015; Freedman, 2017) to small molecules in the Yellow Sea and Bohai Sea (Fig. 6). These compounds may be highly accumulated in the SML.

Under higher surface seawater temperatures in the East China Sea, larger molecules may be largely decomposed to small molecules, TMA and DMA. TMA and DMA were dissolved in bulk seawater with less TMA and DMA enriched in the SML.





Based on the exponential correlation between basic gases and ambient temperature, we inferred that

surface seawater temperature was likely one of the key factors controlling the release of $TMA_{gas}$, $DMA_{gas}$,

and $NH_{3gas}$ from the seas to the atmosphere. The disproportional release of alkaline gases and

corresponding particulate counterparts implied that the enrichment of $TMAH^+$ and $DMAH^+$ in the SML

may be overwhelmingly determined by the release of particulate $TMAH^+$ and $DMAH^+$, although the

extent of enrichment may be largely affected by surface seawater temperature.

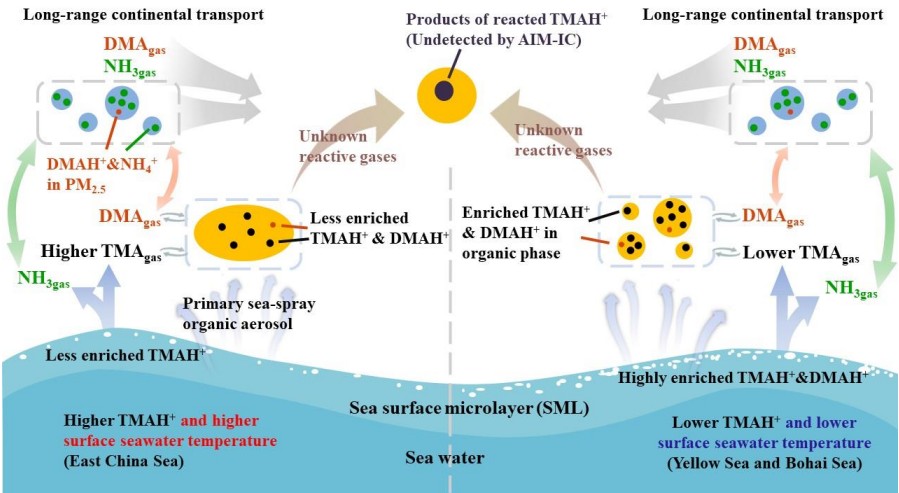

**Figure 6: A schematic illustrating the release of basic gases and their counterparts from the two different
seas and potential atmospheric reactions.**

Combining no correlation between the molar ratios of $TMAH^+$ to $NH_4^+$ in $PM_{2.5}$, the ratios of $TMA_{gas}$

over $NH_{3gas}$, and the data with substantially larger ratios of $TMAH^+$ to $NH_4^+$ compared to those of $TMA_{gas}$

to $NH_{3gas}$, it can be inferred that the observed $TMAH^+$ in the marine atmospheres were probably

overwhelmed by primary sea spray organic aerosols, and existed mainly in either organic phase or mixed

phase. Secondary reactions in the marine atmosphere further led to the conversion of $TMAH^+$ as

chemicals undetectable by AIM-IC, rather than forming new detectable particulate $TMAH^+$.

The sea-derived $DMA_{gas}$ and $NH_{3gas}$ were expected to exhibit an equilibrium with aerosols containing

$NH_4^+$ and $DMAH^+$ from continental transport, but the equilibria were seemingly not achieved over the

three seas. Thermodynamic models, including gas, aqueous phase, organic phase, and mixed phase, are



needed to explain these results (Chan and Chan, 2013; Qiu and Zhang, 2013; Pankow, 2015; Chu and Chan, 2017; van Pinxteren et al., 2019).

The reuse of $NH_4^+$ by phytoplankton may also largely affect the ratios of $DMA_{gas}$ to $NH_{3gas}$ and the ratios of $TMA_{gas}$ to $NH_{3gas}$ in their emissions, which requires further investigation. The extent of degradation

of TMA to DMA in different sea zones may vary significantly, leading to different ratios of $DMAH^+$ to $TMAH^+$ in their primary marine emissions. These factors likely complicated the ratios of $DMA_{gas}$ to $TMA_{gas}$ and $DMA_{gas}$ ($TMA_{gas}$) to $NH_{3gas}$ in their marine emissions and should be considered when estimating their emissions.

In addition, primary particulate $TMAH^+$ and $DMAH^+$ were distributed mainly in submicron atmospheric

particles. Their concentrations generally increased with decreasing particle size. In contrast, the size distribution of secondary particulate $DMAH^+$ should be similar to that of particulate $NH_4^+$ (Xie et al., 2018; Hu et al., 2018). Considering the largely increased ratios of $TMAH^+$ to $NH_4^+$ in <0.43 µm particles, the particles containing $TMAH^+$ may yield contributions comparable with anthropogenic particles to cloud condensation nuclei in less polluted marine atmospheres over the China Marginal Sea.

*Data availability*. The data of this paper are available upon request (contact: Xiaohong Yao, xhyao@ouc.edu.cn).

**Acknowledgment**

This research is supported by the National Key Research and Development Program in China (grant no. 2016YFC0200504), the Natural Science Foundation of China (grant no. 41776086)

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
