# Peer review of "Mapping gaseous amines, ammonia, and their particulate counterparts in marine atmospheres of China's marginal seas: Part 2 - spatiotemporal heterogeneity, causes, and hypothesis"

_Atmospheric Chemistry and Physics, 2021_

## Author Response (AR1)

November 1, 2021

Associate Editor, Dr. Maria Kanakidou

**Response Letter: Revision of Manuscript # acp-2021-301**

Dear Dr. Maria Kanakidou:

We have made revisions according to the comments. Here is a point-by-point summary of our response to comments and suggestions. The responses may be revised according to the final revision of the manuscript in case the language editing may change the ms to some extent. The comments are listed first, and our responses follow each comment. We also checked and revised the whole manuscript and figures.

Best regards!

Sincerely,

Xiaohong Yao, Ph.D.
Ocean University of China

**Response to comments by Anonymous Referee #1**

*This paper presents measurement results of gaseous ammonia, amines and their particulate counterparts in marine atmospheres of China. The work analyzed their concentrations as well as the conversions. The paper is general well written. As amines are found to be increasingly important, this work, in particular, determined the concentrations in gas and particle phase together, provide useful knowledge to understand its role in atmosphere. This reviewer overall recommends its acceptance in ACP, with a few issues listed below to be addressed first:*

*There are many amines in the air, although TMA and DMA might be relatively abundant, but why other amines were not measured here and these two are the most important, this point needs clarification.*

**Response:** In the revision of lines 42-45 and lines 102-108, we added that "The biochemical origins of TMA and DMA in marine atmospheres have been well documented to be released from the degradation of glycine betaine (GBT) and trimethylamine N-oxide (TMAO), which help marine organisms resist the fluctuation of salinity (Lidbury et al., 2014, 2015)." and "Note that mono methylamine cannot be detected by the AIM-IC using the analytical column CS17A (2x250 mm) in this study. The concentrations of triethylamine were generally undetectable so that the data were analyzed here. No biogenic origin in marine environment has been reported for diethylamine, although it may be detected as $TMAH^+$ by the AIM-IC. In 2021, we tried a new analytical column CS20 (2x250 mm) to analyze amines including diethylamine and the unpublished data confirmed its concentration to be negligible relative to $TMAH^+$ in the marine atmosphere of marginal seas of China."

*Section 2: Although the measurement uncertainty or detection limits might have been mentioned in your companion paper, a brief summary with key points can be described here for clarity.*

**Response:** In the revision, we added "The limits of detection (LOD) of $NH_4^+$, $DMAH^+$ and $TMAH^+$ in the atmosphere were 0.0004, 0.004 and 0.002 μg m$^{-3}$." According to the comments of the companion paper, we checked all data and revised the concentrations below the LOD to 1/2 LOD in the revision.

*Section 3.1: What is the PM2.5 levels? It might be better to include it in the figures for comparison*

**Response:** The corresponding mass concentrations of $PM_{2.5}$ were not available in this study. This has been added in the revision of lines 135-136.

*Line 135: It is not very clear to the reviewer, how to reach this conclusion.*

**Response:** We re-organized the paragraph, particularly that we changed the technical

term "actual time emission potential" to "temperature-driven oceanic emission". We hope the revision can make the paragraph more readable.

*Line 186: I am sure this is correct, however, have you tried any back trajectory analysis to confirm it? Why NH4+ concentration can be significantly influenced by continental transport as you mentioned later but not NH3gas?*

**Response:** Agree. The 24-hr air mass backward trajectories at 100 m, 500 m, and 1000 m above sea level has been added in the revised Support Information. The air mass backward trajectories implied that the air masses with high $TMA_{gas}$ and $TMAH^+$ concentrations were from the marine atmosphere. In reverse, the air masses with high $NH_4^+$ concentrations were from the continental atmosphere. The part has been added in the revision of lines 159-161 and lines 163-164.

Since the marine atmosphere is shortage of $SO_2$ and $NO_x$ relative to the upwind continent atmosphere, much less ammonium salts can be formed in the marine atmosphere than in the upwind continent atmosphere. Thus, $NH_4^+$ is better than $NH_{3gas}$ as a tracer of continent outflow. Compared with $NH_4^+$ aerosols, the residence time of $NH_{3gas}$ in the atmosphere was much shorter (Yao and Zhang, 2013). The long-range transport of $NH_{3gas}$ was not comparable with $NH_4^+$ aerosols. The part has been added in the revision of lines 158-159 and lines 209-211.

*Line 187: Is there any recent measurement results of TMAgas or NH3gas for comparison and to confirm the results in this study?*

**Response:** To best our knowledge, no recent measurements of $TMA_{gas}$ and $NH_{3gas}$ in the marine atmosphere were recently reported in the literature. We did find the measurement in the continent atmosphere in Nanjing, China (Zheng et al., 2015). The new data was compared with our measurements in the revision.

*Overall, what are the differences between TMAgas and DMAgas, and TMAH+ and DMAH+, regarding their sources etc? It is not very clear in the conclusion and Figure 6. Can you summarize it?*

**Response:** The part has been added in the revision as "Semi-continuous measurements of concentrations of basic gases and their counterparts over the East China Sea, Yellow Sea, and Bohai Sea showed large spatiotemporal variations. The average concentration of TMAgas was 0.10±0.04 µg m$^{-3}$ over the East China Sea in Campaign B, and decreased by approximately by 70% over the Yellow Sea and the Bohai Sea in Campaigns A and B, with the corresponding TMAgas concentration 0.031±0.009 and 0.037±0.011µg m$^{-3}$. In contrast, the average concentration of $TMAH^+$ in $PM_{2.5}$, over the East China Sea was 0.098±0.068 µg m$^{-3}$, while the average increased by approximately 200% to 0.28±0.18 µg m$^{-3}$ over the Yellow Sea and the Bohai Sea in Campaign A. Comprehensive analysis indicated that both TMAgas and

particulate TMAH+ were released from the seas. The disproportional release of $TMA_{gas}$ and particulate $TMAH^+$ from the East China Sea, compared with the Yellow Sea and the Bohai Sea, however, pointed to a differential enrichment of $TMAH^+$ in the SML.

In Campaign B, the average concentration of $DMA_{gas}$ over the East China Sea was $0.012\pm0.011$ µg m$^{-3}$, and largely decreased to $0.002\pm0.001$ µg m$^{-3}$ over the Yellow Sea and the Bohai Sea. The moderately good correlation between $DMA_{gas}$ and $TMA_{gas}$ suggested that the observed $DMA_{gas}$ was likely derived from marine emissions with $TMA_{gas}$. The average concentration of particulate $DMAH^+$ was $0.019\pm0.014$ µg m$^{-3}$ over the East China Sea in Campaign B. When the concentrations of $NH_4^+$ exceeded 2 µg m$^{-3}$, the corresponding particulate $DMAH^+$ was dominantly from the long-range continental transport. However, the sea-derived $DMAH^+$ was likely to be the major contributor in most of cases when the concentrations of $NH_4^+$ were below 2 µg m$^{-3}$. When the concentrations of $NH_4^+$ were below 1 µg m$^{-3}$, the primary emission of $DMAH^+$ likely contributed to the observed $DMAH^+$ to some extent."

**Response to comments by Anonymous Referee #2**

*This paper is of potential scientific interest and within the scope of ACP since it presents new results on gaseous and particulate amines in China's marginal sea atmosphere. However, the paper is not easy to read and would benefit from being carefully edited in English.*

**Response:** The authors thank the comments and revise accordingly. The original manuscript has been polished using the language-editing Service provided by the Webshop. Unfortunately, the service seems unsatisfiable and we will try other services before the re-submission.

*I would add a table with at least five columns the name of the campaigns or E-periods mentioned in the manuscript, the date and rough location, the average concentration of amines and their potential sources. This will facilitate following the discussion.*

**Response:** Agree. Table 1 has been added.

*In addition, there are conclusions by the authors on the sources of various amines but it remains unclear how these have been reached. On this point, back trajectory analysis could support conclusions on amines origin.*

**Response:** Thanks for the advice. 24-hr air mass backward trajectory analysis at 100m, 500m, and 1000m above sea level has been added in the revision.

*I suggest that the authors address these major concerns and perform the required corrections/rephrasing before a decision is made on this manuscript.*

**Response:** We have revised the manuscript accordingly.

*Specific comments:*

*Line 17, line 123, 162 and at many other places in the text with similar syntax: 'on December 27,2016 – January 6, 2021' I suggest changing to 'from December 27, 2016 to January 6, 2021*

**Response:** Agree. Revised. We have asked the language editor carefully proofreading the type of issues.

*Line 18: of gas phase atmospheric trimethylamine (TMAgas)*

**Response:** Revised.

*Line 21: during the period from 7 to 16 January*

**Response:** Revised.

*Line 22: from 9 to 22 December*

**Response:** Revised.

*Line 24: than the 0.28 …. observed over the Yellow Sea*

**Response:** Revised.

*Line 26: correlation was found over …*

**Response:** Revised.

*Line 30: I suggest removing 'also'*

**Response:** Agree. Revised.

*Line 48: cause*

**Response:** Revised.

*Line 86: short-lived*

**Response:** Revised.

*Line 89: are here referred to… respectively.*

**Response:** Agree. Revised.

*Lines 85-97: could you provide information on the duration of each sampling ?*

**Response:** Each sample was collected for 1 hour, which was added in the revision as "semi-continuously measured the hourly average concentrations of gaseous species of interest, and particulate counterparts, in $PM_{2.5}$".

*line 107: samples were found to reach …*

**Response:** Revised.

*Line 136: it remains unclear to me how this conclusion has been reached*

**Response:** The original parts are indeed not readable. We reorganized the paragraph in the revision and hope that the new version works.

*Line 152: from 15:00 on December 16 to 01:00 on December 19…*

**Response:** Revised.

*Line 158,159: could the authors explain what they mean with ' actual time emission potentials'?*

**Response:** The part has been revised as "temperature-driven oceanic emission". In fact, we reorganized the paragraph in the revision and hope that the new version works.

*Line 163: do you mean 'East China Sea'?*

**Response:** Yes. Revised.

*Line 171: was detected*

**Response:** Revised.

*Line 183: TMAgas, respectively.*

**Response:** Revised.

*Line 199: was observed from 27 to 30 December*

**Response:** Revised.

*Lines 226-229: can you say more about the outliers? Is it possible you lose important information by subtracting them ?*

**Response:** Thanks for the comments. The $DMAH^+$ concentrations of the two of three outliers are 0.069 and 0.040 µg m$^{-3}$, corresponding $NH_4^+$ concentrations are 2.22 and 2.06 µg m$^{-3}$. These two data were measured in the initial few hours of AIM-IC being restarted. The relatively high values were very likely due to incompletely cleaned residuals of $DMAH^+$ in the system, although the residuals of $NH_4^+$ had been completely washed out. Thus, the two data were exhaustively removed in the revision. There was still one outlier of $DMAH^+$ concentration at 0.035 µg m$^{-3}$, with $NH_4^+$ concentration 2.31 µg m$^{-3}$. The outlier is yet to be explained.

*Figure 3: Figure description: explain the inner frame in panel (a). Also in panel (c) move 2nd regression higher to avoid overlapping with open circles.*

**Response:** We added an inner frame in figure 3(c) to avoid overlap and the detailed

description of figure 3 including the inner frame in panel (a) and (c) was revised in the revision.

*Line 234: clearly explain what you mean by 'secondary regression equation'*

**Response:** The expression was indeed confused to readers, we replaced it with the equation "y=0.013x-0.012" in the revision.

*Lines 236-245: This part of the discussion is not clearly written. Please rephrase to clearly explain how you reach the conclusions.*

**Response:** The part has been greatly shortened to make it readable.

*Line 240: 'values' of which amine ?*

**Response:** The values of continental $DMAH^+$ concentration, which was added in the revision.

*Line 249: why 'net'?*

**Response:** The part has been deleted. When the concentrations of $NH_4^+$ were below $1\mu g\ m^{-3}$, the concentration of $DMAH^+$ ranked at a lower level. Too much discussion was unnecessary and thereby removed in the revision.

*Line 250: due to primary emissions*

**Response:** The part has been deleted.

*Line 253: remove 'in contrast'*

**Response:** The part has been deleted.

*Line 256: please add a reference*

**Response:** The part has been deleted.

*Line 266: started at 23:00 on December 27 and ended at 13:00 on December 30*

**Response:** Revised.

*Line 268: use same syntax as above*

**Response:** Revised.

*Line 271:  primary TMAH+ present in PM2.5*

**Response:** Revised.

*Line 271-274: please rephrase the sentence*

**Response:** The part has been revised as "E-period 3 started from 00:00 on 15 December to 11:00 on 19 December 2019 during Campaign A, when either an increase in concentration of $TMA_{gas}$ or particulate $TMAH^+$ was observed without a corresponding increase in their counterparts. The feature was similar to that of E-period 2."

*Line 276: true for all observations ...*

**Response:** Revised.

*Line 281: during E-periods...*

**Response:** Revised.

*Lines 312-313: please rephrase*

**Response:** The part has been revised as "Ratios of aminium over ammonium in bulk surface seawater and/or the SML of the corresponding sea zone may vary to some extent and complicate the observational results."

*Line 334: provide equation*

**Response:** the equation has been added in the Supporting Information.

*Lines 347, 349: spell out to which ratios do you refer?*

**Response:** The ratios are referred to "$TMA_{gas}$ to $NH_{3gas}$", which have been added in the revision.

*Line 350: from ... to ...*

**Response:** Revised.

---

## Author Response (AR2)

Dear Editor

Thank you very much. We have revised the manuscript according to the comments. Here is a point-to-point response to the comments and suggestions.

Your Sincerely.

Xiaohong

Prof. Xiaohong Yao (Ph.D)
Ocean University of China

*Thank you for the revision of your manuscript which has improved its readability. However, there is a number of further corrections to be done before its publication in ACP.*
*These are listed below. Please perform them and submit a revised version for publication in ACP.*

*Outliers: could you comment if the outliers have specific characteristics, sources etc*

**Response:** One data in Figure 3c was subject to the second injection when the AIM-IC was restarted. The first injection has been removed in the data analysis. The outlier could still be due to the incompletely removed residual in the system. Therefore, the data was removed in the revision through the manuscript. The outlier in the remaining outlier was confirmed as the normal signal. However, we can not identify the cause for the signal. The same can be said to the outlier in Figure 3b. This has been clarified in the revision.

*Line 43: well documented with these amines to be released*

**Response:** Revised.

*Line 108 : TMAH+ do you mean DMAH+ ?*

**Response:** We have confirmed it should be $TMAH^+$.

*Line 150: 'The values' do you mean ' TMAgas values'*

**Response:** Yes. Revised.

*Line 156: I think a connecting sentence is missing, something like ' This is supported by the following.'*

**Response:** Added.

*Line 157: replace '1)' by 'First,'*

**Response:** Revised.

*Line 161: from continental input*

**Response:** Revised.

*Line 169: replace '2)' by 'Second,'*

**Response:** Revised.

*Line 175: replace '3)' by ' Third'*

**Response:** Revised.

*Line 213: replace 'was' by ' is'*

**Response:** Revised.

*Line 213: ...'the observed ratios'... please provide average and range*

**Response:** The part has been added in the revision as "the observed ratios of $TMA_{gas}$ to $NH_{3gas}$ ranged from 0.01 to 0.10, with the average value of 0.04±0.01, which were two orders of magnitude larger than those that have been reported in marine atmospheres and adopted for modeling."

*Lines 442, 443, 451: ' in Campaigns' I would rather use ' During campaign'*

**Response:** Revised.